# Tissue Engineering and Regeneration of the Human Hair Follicle in Androgenetic Alopecia: Literature Review

**DOI:** 10.3390/life12010117

**Published:** 2022-01-14

**Authors:** José María Llamas-Molina, Alejandro Carrero-Castaño, Ricardo Ruiz-Villaverde, Antonio Campos

**Affiliations:** 1Department of Dermatology, Hospital Universitario San Cecilio, 18016 Granada, Spain; ricardo.ruiz.villaverde.sspa@juntadeandalucia.es; 2Department of Anatomical Pathology, Hospital Universitario San Cecilio, 18016 Granada, Spain; alejandro.carrero.sspa@juntadeandalucia.es; 3Instituto Biosanitario de Granada, Ibs, 18016 Granada, Spain; acampos@ugr.es; 4Tissue Engineering Group, Department of Histology, Faculty of Medicine, University of Granada, 18016 Granada, Spain

**Keywords:** hair follicle, androgenetic alopecia, tissue engineering, regenerative medicine, bioengineering

## Abstract

Androgenetic alopecia (AGA) is an androgen-dependent process and represents the most frequent non-scarring alopecia. Treatments for AGA do not always achieve a satisfactory result for the patient, and sometimes cause side effects that lead to discontinuation of treatment. AGA therapeutics currently includes topical and oral drugs, as well as follicular unit micro-transplantation techniques. Tissue engineering (TE) is postulated as one of the possible future solutions to the problem and aims to develop fully functional hair follicles that maintain their cyclic rhythm in a physiological manner. However, despite its great potential, reconstitution of fully functional hair follicles is still a challenge to overcome and the knowledge gained of the key processes in hair follicle morphogenesis and biology has not yet been translated into effective replacement therapies in clinical practice. To achieve this, it is necessary to research and develop new approaches, techniques and biomaterials. In this review, present and emerging hair follicle bioengineering strategies are evaluated. The current problems of these bioengineering techniques are discussed, as well as the advantages and disadvantages, and the future prospects for the field of TE and successful hair follicle regeneration.

## 1. Introduction

Hair is a primary characteristic of mammals, and exerts a wide range of functions including thermoregulation, physical protection, sensory activity and social interactions [1]. Hair loss or alopecia can be a clinical manifestation of a large number of pathologies, as well as an androgen-dependent process in the case of androgenetic alopecia (AGA). The loss of hair follicles (HFs) caused by injuries or pathologies may affect patients’ psychological well-being and social aspects of some persons; likewise, in certain cases it may endanger the inherent functions of the skin [2]. AGA affects both males and females and is characterized by a non-scarring progressive miniaturization of the HF and loss of terminal hair with a characteristic pattern of alopecia which varies slightly according to gender. Its prevalence increases with age and it has been estimated that half of the male population will experience hair loss by the time they reach fifty [3]. Nowadays, the economic outlook for AGA therapeutics is a global annual market valued at approximately 4 billion U.S. dollars, with a growth rate of about 2%. Current treatments may vary depending on the type of alopecia and patient preferences, involving topical, oral and surgical treatments. However, topical and oral approaches do not always achieve an optimal result and sometimes cause side effects that may even lead to discontinuation of treatment [4]. To date, Minoxidil and Finasteride are the two treatments for AGA that have been approved by the US Food and Drug Administration (FDA) and the European Medicines Agency (EMA). Topical Minoxidil is available as a 2% or 5% solution and foam and only 40% of patients experience cosmetically significant improvement. Moreover, contact irritant dermatitis and facial hypertrichosis are common adverse effects. Regarding Finasteride, its approved dose of 1 mg daily prevents the conversion of testosterone to dihydrotestosterone by inhibiting the enzyme 5-alpha reductase type 2. Finasteride stops hair loss in over 95% of men, in 66% achieves moderate hair regrowth and in only 5% marked hair regrowth. Side effects are reversible upon discontinuation of treatment and mainly affect the sexual domain, with decreased libido and erectile dysfunction being the most frequent side effects [5]. Hair restoration surgery (autologous hair transplant) is today the most effective method in advanced AGA, even though this technique is not without drawbacks. In addition, the lack of follicles in the donor area, as occurs in some subjects, may represent its main limitation [6].

Major advances in the treatment of alopecia are expected in the forthcoming years. The search for alternative therapeutic solutions capable of generating an unlimited number of HFs ex novo has been encouraged by the limited efficacy and possible side effects of the current treatments [4]. The knowledge of hair biology is improving and new signaling pathways and organogenesis processes are being discovered which have the potential to produce more effective therapeutic modalities [3]. Besides this, as in many other areas of medicine, tissue engineering (TE) has been postulated as a future alternative to the problem posed. TE represents the union between engineering and health sciences to create artificial tissue substitutes. This discipline, together with cell therapy and gene therapy, constitute the three main fields of the so-called ‘advanced medical therapies’. There are different approaches within TE: Substitution of cells capable of performing the function of damaged cells; use of inductive signals and use of cells on matrices made of collagen or synthetic polymers [7]. HF bioengineering approaches are grounded on the accumulated knowledge on reciprocal epithelial-mesenchymal (EM) interactions controlling embryonic organogenesis and postnatal HF cyclic growth [4]. However, the reconstitution of a fully organized and functional HF from dissociated cells originated under defined tissue culture conditions is an unresolved challenge and the notable improvements in cognizance of key processes in HF biology have not yet translated into replacement therapies in clinical practice [8]. For this reason, it becomes necessary to investigate and to develop new procedures and new biomaterials that allow the generation of HF and its successful implementation and maintenance in vivo.

## 2. The Hair Follicle: Biology and Embryology

Skin layers include epidermis and dermis. The first one serves as a barrier preventing interior losses in addition to injuries due to external aggressions, and dermis provides circulation and nutrition. Other functions performed by cells in the skin include immune recognition and memory, ability to repair injuries, thermal regulation and communication. Basal membrane is a specialized aggregation of binding molecules which allows the proper union between epidermis and dermis. Skin appendages, including HFs, sebaceous glands and sweat glands, are linked to the epidermis but project deep into the dermal layer [9]. 

The HF is the part of the skin on which one or more hair structures develop and are nourished (Figure 1). In addition to generating hair with the aforementioned functions, HFs also serve as anchors for sensory neurons, arrector pili muscles and blood vessels. The HF is the most dynamic skin structure and one of the most active in the whole organism, and forms a complex, multi-cylindrical organ with sebaceous, apocrine glands and the arrector pili muscle [1,9]

The evolution of an HF over time is known as the hair cycle. The follicular cycle varies according to the body region and even in the same area. Therefore, this cycle is specific to each follicle and independent of the others. It is estimated that around 13% of hair is in the telogen phase and 1% in catagen and exogen, and at this point the hair shaft is detached. Thus, as the human scalp has an average of 100,000 and 150,000 hairs, between 50 and 100 hairs are shed per day. 

The hair cycle begins with an active growth phase known as ‘anagen’ in which matrix forms a new hair. The nutrition of HF from the blood supply allows its growth and maintenance. Wnt, activin/BMP and TGF-β/BMP3 represent the main signalling pathways. In the ‘catagen’ stage, lasting 1 to 2 weeks, the deepest portion of the HF begins to collapse and the blood supply ceases. Molecules that promote the transition to catagen include the growth factors FGF5 and EGF, neurotrophins such as BDNF and others as p75-neurotrophin receptor, p53 and TGFβ-family pathway members such as TGFβ1 and the BMPRIa. Catagen is followed by the ‘telogen’ phase, with a duration of 5 to 6 weeks, in which the papilla cells are already completely separated from the HF. The telogen stage is traditionally called the ‘rest’ phase, due to inhibitory signals (e.g., FGF-18, BMP-6). The ‘exogen’ phase results in the shedding of the hair. Afterwards, the HF begins a new phase of anagen and the old hair is replaced by a new one [9,10]. In AGA, the follicle miniaturization is accompanied by a decrease of the percentage of HFs in anagen and an increase in the telogen [11]. Versican is a chondroitin sulfate proteoglycan and a component of the extracellular matrix and is known to play a role in immunity and inflammation [12]. The dermal papilla (DP) expresses versican mostly in the anagen stage, with a decreased expression in the catagen stage, and is nondetectable in telogen, indicating its importance in the process of maintaining normal HF growth. Patients affected by male AGA were found to have little-to-no versican expression in the DP [13,14]. 

HF morphogenesis begins towards the end of the first trimester of pregnancy through complex morphogenetic processes resulting from highly coordinated series of bidirectional EM interactions. The Wnt signaling pathway is considered to be the main regulator. HF morphogenesis includes three main stages: Hair placode formation, HF organogenesis and cytodifferentiation [15]. Hair development begins through inductive signals from the undifferentiated epithelium leading to the accumulation of β-catenin in the superficial dermis through the Wnt pathway. The dermal cells respond with the secretion of morphogens, such as Wnt, ectodysplasin A (EDA) and fibroblast growth factors (FGF), inducing the invagination of epidermal cells and formation of the epidermal placode. Subsequently, the placode matures into the primary hair germ initiating organogenesis. The primary hair germ secretes platelet-derived growth factor A (PDGF-A), sonic hedgehog (SHH) and FGF-20, which promote dermal cell condensation. The signs of dermal condensate include Wnt, SHH and hepatocyte growth factor (HGF); this initiates the downward growth of the hair germ to form the bulbous protrusion, which envelops the condensate and reinforces the interaction interface. Various signalling molecules are related to the cytodifferentiation, which is characterized by development of all the compartments of the HF [9,10,16,17].

## 3. Stem Cells Populations of the Hair Follicle 

The functional and cyclic activities of the HF are based on coordinated communication between different cell populations of epithelial, mesenchymal and neural crest origin [18]. Understanding the anatomy of the HF and the stem cell populations operating during postnatal cyclic regeneration is assumed to be crucial for a TE-based solution to the problem. The HF is composed of both epithelial and mesenchymal elements. The first group includes epithelial cells organized in the form of three concentric layers, which from more external to internal are the outer root sheath (ORS), the inner root sheath (IRS) and the hair shaft (HS) [19] (Figure 1). In the ORS, a unilateral eccentric thickening is found at the insertion point of the arrector pili muscle. This area is known as the bulge and in this region the largest population of follicular stem cells (SCs) settle. SCs are a type of cell with self-renewal and multilineage differentiation capacities [20]. Their location is known as ‘niche’, a place in which specific conditions and microenvironment are maintained to support them, and it varies from one population of SCs to another [21]. The HF represents the main reservoir of cutaneous SCs [22]. Bulge’s SCs, also known as outer root sheath cells or hair follicle stem cells (HFSCs), have a multipotentencial behaviour by being able to differentiate in several different cell types [23]. Due to this, HFSCs can regenerate the epidermis and at least parts of the HF. HFSCs are expected to have a high regenerative potential and are positive for the stem-cell marker CD34, as well as keratin 15 (K15), leucine-rich G-protein-coupled receptor 5 (Lgr5) and integrin α6 [20,23,24].

In addition to bulge HFSCs, there are other epidermal SC populations in the HF, such as neuronal progenitor cells, sweat gland stroma-derived SCs, sebaceous gland SCs and melanocyte stem cells (MeSCs). There are also SCs located in the isthmus region and in the infundibulum. SCs of the isthmus express MTS24 and Lgr6, while SCs of the infundibulum are Lrig1-positive. The former help maintain the sebaceous gland and the interfollicular epidermis, while the latter contribute also to the upper pilosebaceous unit (Table 1) [20,25,26,27]. Label retaining cells (LRCs) are SCs which retain the chromatin label over considerably longer time periods than the surrounding cells in the tissue [22]. In the HF, the bulge region hosts the majority of the LRCs. It has been shown that these cells are able to proliferate under physical injury or certain stress conditions [22,26].

Each HF is placed on a DP composed of a set of dermal papilla cells (DPCs) of mesenchymal origin. The connective tissue sheath or dermal sheath (DS) is the other mesenchymal element, which lies between the ORS and the papillary dermis. DPCs exhibit inductive properties, playing an important role in the follicular cycle and hair growth [25]. These cells can express the dermal fibroblast markers PDGFRα, type I collagen, vimentin, fibronectin and fibroblast-specific antigens [28]. The study by Rahmani et al. [29] suggests that the DP plays a fundamental role in the restoration of hair growth after damage, and its harm in the case of certain traumas, diseases or in old age would therefore entail the disturbance in the HF cycle. These authors eliminated SCs from the DP and observed a delay in the regeneration of the HF, and an alteration in the specification of hair type. 

During the anagen phase, DP stimulatory signalling overcomes the threshold imposed by the inhibitory bulge microenvironment and the quiescent HFSCs are activated. Thus, HFSCs proliferate and generate HF-transit amplifying cells (HF-TACs). HF-TACs allow the growth of the HF through differentiation into eight distinct epithelial lineages and sebaceous glands. This process will eventually result in a mature HF. When the cells of the matrix exhaust their proliferative capacity, hair growth stops and the follicle begins the catagen phase, which leads to degeneration of the lower 2/3 of the follicle, while the bulge zone remains intact. When injury occurs, HFSCs can also differentiate into cells of the interfollicular epithelium and sebaceous gland [8,25,30]. 

Liu JY et al. [31] confirmed that DPCs and the DS cells from human HFs express the mesenchymal stem cell immunophenotype and possess multi-lineage differentiation potential; therefore, they were named human hair follicle-derived mesenchymal stem cells (hHF-MSCs). Outside their usual location, hHF-MSCs have the capacity to differentiate into osteogenic, myogenic, adipogenic and even hematopoietic lineages [9]. Accordingly, the HF may be a readily accessible source of autologous human MSCs that can be used for TE and regenerative medicine [29].

Regarding SCs and AGA, infiltrating lymphocytes and mast cells have been identified around the follicle, especially in the bulge area [32]. Garza L. A. et al. [33] demonstrated that this type of alopecia is characterized by a defect in the conversion of HFSCs into active progenitor cells, and not in a loss of thereof. By flow cytometry, these authors quantified the markers of HFSC as K15, CD200, CD34 and integrin α6. K15 stem cells were maintained in bald scalp samples, and high levels of K15 expression correlated with a small SC size and quiescence. However, CD200, integrin α6 and CD34 cell populations were diminished.

## 4. Bioengineering Strategies for Human HF Regeneration

Given the potential of HF cloning, regenerative medicine in the treatment of AGA focuses on the production of instructive germs of HF cells expanded in vitro to generate fully functional HFs after transplantation into the patient’s bald scalp [8]. HF regenerative medicine should focus on the EM interactions that are continuously occurring between its various components. The inductive signalling is issued primarily from the mesenchymal (dermal) component being received by the epithelial cells. Because of this, any TE approach of the HF generation under in vitro conditions will require (Figure 2): isolation of epithelial and DPC populations, expansion of them by culture, maintenance of their proliferative and inductive properties, and the provision of exogenous signals to provide correct EM interactions [34]. Finally, the trichogenic ability should be assessed in an in vivo model. Several studies [35,36] have proven how isolated murine bulge cells when combined with neonatal DPCs reform the entire cutaneous epithelium, including HF, epidermis and sebaceous gland. 

HF is considered an immune-privileged site, as it does not express MHC class I antigens [37]. In 1999, A. J. Reynolds et al. [38] proved that an allogenic approach in the bioengineering of the human HF is possible. These authors induced growth of HF cells by transplantation of microdissected DP and DS into an incompatible host of the other sex without rejection. Nevertheless, the current TE approach for regeneration of HFs in AGA is mainly autologous. For this purpose, both follicular and non-follicular TE methods will require an initial isolation of progenitor cells and posterior expansion in culture. In the case of non-follicular techniques, the next step will be appropriate differentiation to follicle mesenchymal and epithelial SCs. The final step in both approaches is to combine the epithelial and mesenchymal components and form engineered instructive minibulbs capable of generating mature and functional HFs once transplanted into the recipient’s skin (Figure 2) [8]. In addition, the aims of generating HFs through TE also include their application in the research and clinical field, such as their use in hair transplantation procedures and pharmacological tests [13].

### 4.1. Follicular Based Approaches

#### 4.1.1. Dermal Papilla Cells (DPCs)

##### Isolation

In a follicular approach, the first step consists of isolating the DPCs and the HFSCs. For this purpose, HFs should be extracted from the donor area (part of the scalp not affected by this type of alopecia) by punch biopsies or follicular unit extraction (FUE) in order to obtain these cells for a subsequent culture. Regarding DP and DPCs, enzymatic digestion followed by fluorescence activated cell sorting (FACS-sorting) has been used for the isolation of the DP in rodents. In contrast, robust cell surface markers for FACS-sorting of human DP or DS cells remain to be defined and the extracellular matrix composition found in human DP renders enzymes such as trypsin and collagenase insufficient for digestion of the DP. In addition, enzymatic digestion of collagen IV and fibronectin may affect the intrinsic qualities of the extracellular matrix and the microenvironment it generates, leading to an alteration in the hair-inducing properties. Thus, enzymatic isolation techniques should be avoided in humans and to grow human DPCs in vitro the papilla has to first be isolated via a micro-dissection approach from the follicle into a single cell suspension [8,39,40,41,42]. Methods for the isolation of DPCs are summarized in Table 2.

##### Culture

In 1984, Messenger successfully grew human DPCs [47]. Previously, Jahoda and Oliver first achieved in vitro cultivation of microdissected rat vibrissa DPCs [48]. The specific and necessary microenvironment of the HF, the lack of efficient isolation methods and the loss of innate characteristics in vitro have traditionally posed significant problems in an in vitro expansion of the human DPCs [49]. One of the methods used is the co-culture of DPCs with keratinocytes or keratinocytes-conditioned medium with the aim of maintaining the signals that enable EM interactions during the hair cycle. Modulation of the JAK-STAT signalling pathway has also been shown to increase the inductive capacity of DPC cultures [50]. The problem with 2D monolayer HF models is that they do not reproduce several key features of the HF microenvironment and DPCs rapidly lose their inductivity in culture [13,39]. Probably the most effective culture technique to reproduce the native trichogenicity of DP is the establishment of three-dimensional (3D) sphere cultures [8,51]. Different 3D spheroid models have been developed, and most of them are primarily aimed at creating a microenvironment that allows for the EM interactions that occur in vivo [52]. Three-dimensional Spheroid culture helps DPCs to aggregate and engage in cell–cell contacts, a crucial feature of HF in vivo [39]. Thus, one should try to obtain in vitro a cell structural arrangement as close as possible to the in vivo reality, with seeding of DPCs in wells being very useful [13]. Three-dimensional cultures have been shown to display expression of specific markers of HF induction, unlike 2D cultures [13]. Recently, the different gene expression in 3D spheroids compared to 2D cultures has been investigated in Sika deer (*Cervus nippon*) DPCs [53]. Thus, it was found that versican and CD133 were much higher in 3D cultures compared to 2D cultures, and both are known to be two important drivers of HF-inducing capacity in vivo. The same was true for alkaline phosphatase activity, which is closely related to DP trichogenicity [53]. Higgins et al. [51] demonstrated that 3D cultured human DPC spheroids are capable of inducing de novo HF in intact human recipient skin. Besides this, studies have been published investigating the association with biomaterials such as Polyvinyl alcohol as a scaffold support [54] and even with human placental extracellular matrix hydrogels [55]. Furthermore, the effect of molecular signalling, such as the combination of pharmacological activation of Wnt signalling of DPCs with 3D spheroid culture, has also been evaluated [56]. 

#### 4.1.2. Hair Follicle Stem Cells (HFSCs)

Bulge HFSCs are multipotent cells and represent the largest stem cell population. They actively contribute to HF regeneration in vivo by replenishing different epithelial lineages [23]. In addition, it has been reported that HFSCs are prominently involved in HF neogenesis and repair in vivo and their ablation in the bulge area leads to complete loss of HFs in vivo. Thus, HFSCs as well as DPCs, are required for HF organogenesis and maintenance and should be supplemented in in vitro systems [57]. CD34 and integrin α6 are some of the most widely used markers usually used together to identify HFSCs [24]. FACS-sorting has also been used to identify and isolate these cells, as well as magnetic activated cell sorting and mechanical isolation techniques [24]. 

### 4.2. Non-Follicular Cell Sources for HF Bioengineering 

The non-follicular approach involves those TE techniques that aim to regenerate the HF from SCs obtained outside the follicle [8]. Therefore, in both rodent and human samples, mesenchymal precursors (e.g., skin derived precursors (SKPs) [58], adult dermal fibroblasts [59]) and epithelial precursors (e.g., adult epidermal keratinocytes [60], adult epidermal stem cells [61]) have been used for this purpose. While HF and sebaceous gland reconstitution has been achieved in vivo in mice with spheroid cultures containing epidermal stem cells and SKPs in a hydrogel [61], in humans doubts remain about the potential for inducing these populations and their difficult isolation from the human dermis. 

The drawbacks related to the isolation of precursors mentioned above, as well as the scarcity of cell population, favour the employment of non-autologous reprogramed pluripotent stem cells with multi-lineage differentiation ability obtained from non-hair cell sources [8]. From pluripotent stem cells, such as human embryonic stem cells (hESCs) and human-induced pluripotent stem cells (hiPSCs), specific somatic cell lineages can be obtained. To do so, cytokines that mimic the patterning and positioning signals during embryogenesis have to be used [62,63]. hiPSCs cells are obtained from adult somatic cells that are reprogrammed to overcome this initial limited differentiation potential [64]. Both MSCs and HFSCs can be induced by hiPSCs. hiPSCs may also generate an inductive dermal and receptive epidermal cell population, enabling EM communication in 3D integumentary organ system in vitro [62]. Moreover, Lee J et al. [65] recently succeeded in generating full-form hairy skins derived from iPSCs based on an organoid culture system. However, the generalization of the use of iPSCs for therapeutic purposes in humans has yet to resolve doubts regarding their safety, such as the case of viral integration in the genome and the risk of teratoma formation [8]. Gnedeva K et al. [66] led hESCs to differentiate into neural crest cells, which were subsequently induced into hair-inducing DP-like cells in culture that resulted in HF formation when transplanted under the skin of immunodeficient nude mice.

## 5. Current Challenges and Future Perspectives in Regenerative Medicine Therapy for Hair Loss

New topical treatments which are being investigated in AGA, such as Bimatoprost or Setipripant, have been shown to be effective but not superior to topical Minoxidil [67]. The growing knowledge about the importance of the Wnt pathway in the regulation of the hair cycle means that this pathway is being investigated for the development of new treatments. Some molecules in development, such as SM04554 [68], are being studied with some preliminary evidence of efficacy although they have not yet been compared to Minoxidil. While there is some inflammatory processes in the lower part of the HF and a variation of the hair microbiome in AGA patients with regard to controls, there is no evidence yet to support a treatment aimed at modifying the microbiome of these patients [69]. It has also been suggested that platelet-rich plasma (PRP) treatment promotes hair growth, promotes cell survival and proliferation and prolongs the anagen phase of the hair cycle. In patients with lower grade of alopecia, PRP has demonstrated favourable outcomes when compared with placebo [70].

The problems related to the current treatment of AGA, as well as those recently commented on concerning the upcoming treatments, make regenerative medicine and bioengineering acquire a future relevant role in this field. To date there is no cell therapy product for hair regeneration despite its great potential. This lack is mainly attributed to the still existing problems that hinder the functionality of cultured human hair cells. Moreover, drastic transcriptional changes occur in culture, reducing our ability to understand the role of the papilla in hair growth and cycling with in vitro models [13,39]. Therefore, most studies have focused on the development of 3D-like structures that most closely resemble in vivo HFs in terms of structure, signalling pathways and cycling [13]. Isolation methods also need to be improved because more intact human DPs would allow for more ex vivo analyses [39]. 

While satisfactory results are being achieved in the HF regenerative medicine, there still seem to be a number of important limitations to its clinical implementation. The prosperous large-scale production of functional HFs, their successful in vivo implementation and long-term maintenance, and the aesthetics results, are currently a challenge and an unknown to overcome. Abaci et al. [52] achieved a remarkable attainment, creating ex vivo functional human skin incorporating HFs, and growing these human HFs by implanting them into immunodeficient mice.

In the field of cell therapy, Gentile P. et al. [71] in 2017 obtained HFs from patients with AGA (3-5 in Noorwood-Hamilton classification scale) by punch biopsy and subsequently centrifuged the sample and isolated the HFSCs by a promising CE-certified medical device called Rigeneracons^®^. Afterwards, these cells were not cultured, but were implanted directly in the form of an autologous suspension in some of the patient’s affected areas by AGA, injecting placebo in the others. On average, in treated patients hair count and hair density increased over baseline values 23 weeks after the last treatment with HFSCs. Thus, these authors recommend more studies and clinical trials to follow this first achievement. Moreover, with the Rigenera^®^ device, Zanzottera et al. [72] prepared autologous adipose derived stem cells obtained during hair transplant. In three patients who had received hair transplantation, this autologous suspension was injected in the grafted areas and showed a greater healing of the wounds produced by the hair transplantation and an increase in hair density. To conclude with cellular therapy techniques, recently the hair regrowth with micrografts containing human hair follicle mesenchymal stem cells (hHF-MSCs) was also compared with a saline solution, with better results for the first group when studied by trichogram [73]. Recently, Ruiz et al. also assessed the possible therapeutic effect of autologous micrografts in patients affected by AGA (n = 100). For this purpose, they performed three punch biopsies on the scalp of each patient, which were introduced into the Rigeneracons^®^ medical device to be disaggregated by adding 1.5 mL of sterile physiological solution. The subsequent micrografts were infiltrated into the scalp of the patients. The results were evaluated 4, 6 and 12 months after the application of the micrografts and in all evaluations a significant improvement in hair growth and density, measured by trichogram, was observed [74].

In relation to gene therapy, four mRNAs overexpressed in AGA were identified, and the use of small interfering RNA using biodegradable cationized gelatin microspheres was successful in murine models [75].

Lastly, another drawback is the high economic cost. The establishment of a personalized therapy for the patient will necessarily make it very expensive. The current economic cost of the process also makes its implementation in clinical practice unfeasible, as well as the costs to be borne by the patient [8].

## 6. Conclusions

TE is postulated as one of the future alternatives to the problem of AGA. HFSCs and DPCs have particular appeal for this problem, and their use could also be extended to other areas of medicine. Despite significant advances in bioengineering techniques for HF, especially in murine models, the main drawbacks at present are their high cost, complexity and the difficulty of translating them into clinical practice. If these problems are overcome in the coming years, there is no doubt that bioengineering could revolutionise the current treatment of AGA and even other types of non-scarring alopecia.

## Figures and Tables

**Figure 1 life-12-00117-f001:**
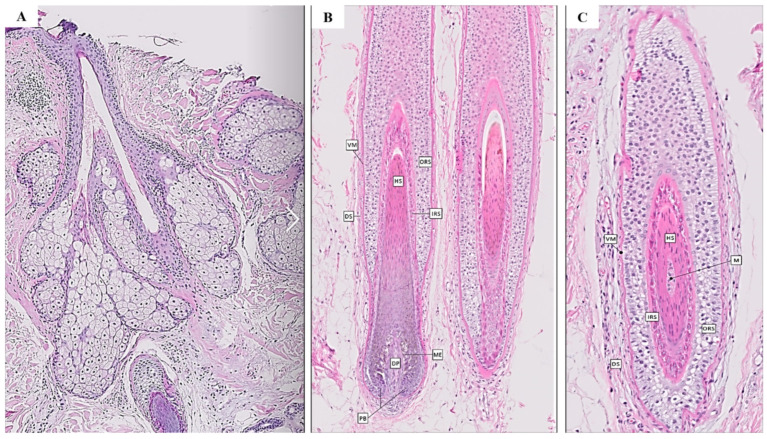
Histological images of the HF. (**A**) Sebaceous glands and their duct opening into hair follicle; (**B**) Vertical; (**C**) Horizontal. **M**: Medulla. **HS**: Hair shaft. **IRS**: Inner root sheath. **ORS**: Outer root sheath. **VM**: Hyaline membrane. **DS**: Dermal sheath. **ME**: Melanocytes. **DP**: Dermal papilla. **PB**: Pilous bulb.

**Figure 2 life-12-00117-f002:**
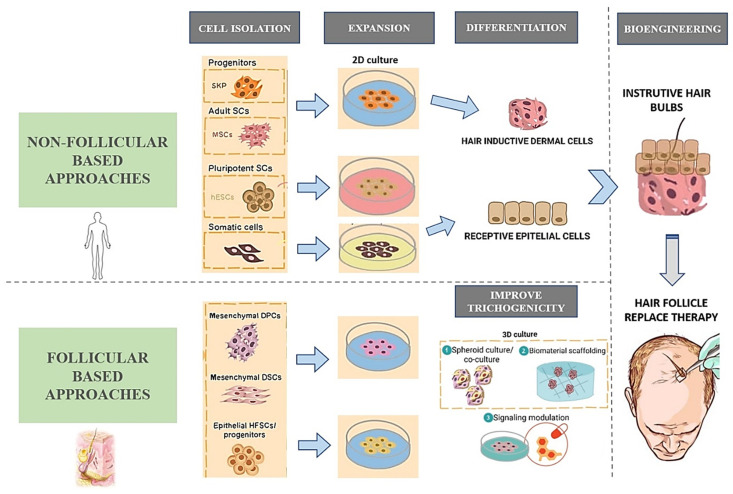
Different approaches to obtain HFs by TE. Adapted from the article ‘Castro, A. R. and Logarinho, E. Tissue engineering strategies for human hair follicle regeneration: How far from a hairy goal?’ First, cell populations are harvested to form the epidermal and dermal components of the HF. If cells are obtained from sites other than the HF, they need to be differentiated appropriately. After initial expansion in culture, trichogenicity will be increased by 3D spheroid culture, biomaterials or by modulation of signalling. Finally, once the HFs have been obtained by follicular or non-follicular techniques, the final step is their implantation and the evaluation of clinical success.

**Table 1 life-12-00117-t001:** Overview of the main stem cell populations located in the hair follicle.

Stem Cell	Location	Main Markers	Origen
Interfollicular epidermal stem cells	Epidermis	Integrin α6 Keratin 5 (K5) Keratin 14 (K14)	Epidermal
Hair Follicle Stem Cells (HFSCs)	Bulge	CD34 Keratin 15 (K15) Leucine-rich G-protein-coupled receptor 5 (Lgr5) Integrin α6	Epidermal
Stem cells of the Isthmus	Isthmus	MTS24 Leucine-rich G-protein-coupled receptor 6 (Lgr6)	Epidermal
Stem cells of the Infundibulum	Infundibulum	Leucine-rich repeats and immunoglobulin-like domains protein 1 (Lrig1)	Epidermal
Dermal Papilla Cells (Hair follicle-derived mesenchymal stem cells)	Dermal Papilla	Nestin Vimentin Fibronectin Sca-1 Markers for fibroblasts, such as collagen I	Mesenchymal

**Table 2 life-12-00117-t002:** Some of the reported methods for the isolation of DPCs [39,42,43,44,45,46].

Isolation Method	Species	Procedure	Advantages	Disadvantages
**Enzymatic dissociation** [43]	Mice, humans	Collagenase treatment of the proximal portion of HF; stop the reaction when DS is digested	Faster and less labour-intensive	Loss hair inductive properties in humans
**Surgical micro-dissection** [42,44,45]	Mice, rats, humans	Dissect bulb of HF; cut the DS to expose the DP	Preserves the intact DP	Labour-intensive
**Inversion technique** [39]	Humans	Variant of a surgical micro-dissection. A fine needle is used to invert the collagen capsule structure of the terminal bulb to the DP	Preserves the intact DP	Labour-intensive
**FACS-Sorting** [46]	Mice	Sort FDP cells with fluoresce labelled surface markers (e.g., CD133 in mice)	Efficient with high purity	No proper markers for human DP have been identified

## Data Availability

No new data were created or analyzed in this study. Data sharing is not applicable to this article.

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
