# Peer review of "Tissue Engineering and Regeneration of the Human Hair Follicle in Androgenetic Alopecia: Literature Review"

_life, 2022, doi:10.3390/life12010117_

Round 1

Reviewer 1 Report

The paper reports the actual research on Tissue Engineering and Regeneration of the Human Hair Follicle in Androgenetic Alopecia: in my opinion is well documented and above all realistic. Please put more informations on Fig. 2 legend, such as abbreviations. 

Author Response

Thank you for your prompt and kind reply. A new cover letter is sent to inform and justify the modifications made by the authors to the article following the reviewers' suggestions. Modifications that have been made are shown in red in document “Edited after major revision (changes in red). Tissue Engineering and AGA. Life Journal. Llamas-Molina et al.”. 

  1. Figure 2 has been delated and changed by a table summarising in a simple way the main characteristics of the main hair follicle stem cell populations. This table will be referred to as Table 1, and the previous table so named will be listed as “Table 2” to follow the order of the text.
  2. The image previously referred to as Figure 3 has again been realised and is now referred as “Figure 2”.
  3. The description in image previously referred to as Figure 3 (actual “Figure 2”) has been scaled down and summarised to make it easier to read quickly when interpreting the image.

Reviewer 2 Report

It can be seen that the author has read and collected many papers related to hair. However, the author contains too much content in one paper, so the content is too comprehensive.

Regeneration of human hair follicles using human-derived cells is not an easy task. Two types of cells (mesenchymal-derived cells and epithelium-derived cells) are essential for hair regeneration. As the author said, among the cells derived from mesenchymal, dermal papilla cells have been reported to regenerate hair in various ways, but cells derived from epithelium have not been reported except for the method of inducing iPSCs.

The author should include a lot of human content for androgenetic alopecia as stated in the title. There are many contents that have been confirmed using mouse cells in many contents. And since the author tries to include too much content in the paper, a lot of unnecessary content is included. Also, there are many expressions of other terms than the terms used by hair researchers.

1. There are a lot of unused terms in the hair field, which confuses me. Also, the abbreviation is expressed incorrectly.

ex) Figure 1. Histological images of the HF.

1a. Transversal → horizontal

1b. Sagittal → vertical

TP: Hair shaft. → HS

RI: Inner root sheath → IRS

RE: Outer root sheath → ORS

FE: Dermal sheath. → DS

PD: Dermal papilla → DP

2.Too many and unnecessary places are marked with (-).

ex) page 1

Hair is a primary characteristic of mammals, and exerts a wide range of functions including thermoregula-tion, physical protection, sensory activity and social interac-tions1. Hair loss or alopecia can be a clinical mani-festation of a large number of patholo-4 gies, as well as an androgen-dependent process in the case of androge-netic alopecia (AGA). The loss of hair follicles (HFs) caused by injuries or pathologies may affects the pa-tients’ psychological well-being and social aspects of some persons, likewise in certain 7 cases it may endan-gers the inherent functions of the skin [2]. AGA affects both males and females and is characterized by a non-scarring progressive miniaturization of the HF and loss of terminal hair with a characteristic pattern of alope-cia which varies slightly accord-ing to gender. Its prevalence increases with age and it has been estimated that half of the male population will experience hair loss by the time they reach fifty [3]. Nowadays, the economic outlook for AGA therapeutics is a global annual market valued at approxi- mately 4 billion U.S. dollars, with a growth rate of about 2%. Current treatments may vary depending on the type of alopecia and patient prefer-ences, involving topical, oral and

3. A lot of the content was taken from other reviews and organized. Authors should cite the original paper as a reference whenever possible.

4. Contents that are not related to the title are mentioned.

- I wonder why terminal hair and vellus hair suddenly appear while explaining hair biology on page 2.

- On page 4, the stem cell population section contains information about hair structure, and hair cycling and hair morphogenesis are mentioned together.

5. in the abstract part “some-times cause side effects that lead to discontinuation”

what do you mean to discontinuation?

6. page 3

“In AGA, the follicle miniaturization is accompanied by a decrease of the percentage of HFs in anagen and an increase in the telogen“.

Versican is only mentioned while explaining the miniaturization part of AGA. There are many various factors, and since versican is expressed only in DP, expression is inevitably weakened during the telogen period when the size of DP is greatly reduced. Also, I wonder why this is included in the hair cycle section.

7. Reference 18 is not indicated in the written text.

8. (Figure 1) and (figure 2) are marked differently.

9. The figure shown in Figure 2 and the content mentioned in the text are slightly different.

“SCs of the isthmus express MTS24 and Lgr6 while SCs of the infundibulum are Lrig1-positive”

10. page 5 “dermal cells (DPCs) of mesenchymal origin”

   Dermal cells are not abbreviated as DPC.

11. page 6 “Bioengineering strategies for human HF regeneration”

 Lack of strategy content about human cells

12.  page 7

4.1.2. Hair Follicle Stem Cells (HFSCs)

Only mouse cells are mentioned.

Author Response

Thank you for your prompt reply. A new cover letter is sent to inform and justify the modifications made by the authors to the article following the reviewer's suggestions. 

  1. The terms in Figure 1 (Histological images of the HF) have been modified.

1a. Transversal → horizontal

1b. Sagittal → vertical

TP: Hair shaft. → HS

RI: Inner root sheath → IRS

RE: Outer root sheath → ORS

FE: Dermal sheath. → DS

PD: Dermal papilla → DP

  1. Figure 2 has been delated and changed by a table summarising in a simple way the main characteristics of the main hair follicle stem cell populations. This table will be referred to as Table 1, and the previous table so named will be listed as “Table 2” to follow the order of the text.
  2. The image previously referred to as Figure 3 has again been realised and is now referred as “Figure 2”.
  3. The description in image previously referred to as Figure 3 (actual “Figure 2”) has been scaled down and summarised to make it easier to read quickly when interpreting the image.
  4. In section 2, the text referring to vellus and terminal hair has been deleted.
  5. In section 3, the term “papilla” has been added for the correct abbreviation: dermal papilla cells (DPCs).
  6. In abstract, the term "discontinuation" has been substituted by “discontinuation of treatment”.
  7. Reference number 18 (Hardy, M. H. The secret life of the hair follicle. Trends Genet. 8, 55–61 (1992) had already been cited in the text. Section 3, second line.
  8. In section 4, sentence “HF is considered an immune-privileged site, as it does not express MHC class I antigens36” has been referenced as its original article and not as the review where it was found. (Westgate GE, Craggs RI, Gibson WT. Immune privilege in hair growth. J Invest Dermatol. 1991 Sep;97(3):417-20).
  9. In section 4.1 (culture), sentence “3D Spheroid culture helps DPCs to aggregate and engage in cell-cell contacts, a crucial feature of HF in vivo38” has been referenced as its original article and not as the review where it was found (Topouzi, H., Logan, N. J., Williams, G. & Higgins, C. A. Methods for the isolation and 3D culture of dermal papilla cells from human hair follicles. Dermatol. 26, 491–496 (2017).
  10. Answer to comment of revisor: “The stem cell population section contains information about hair structure, and hair cycling and hair morphogenesis”. The authors consider that include information about hair structure, hair cycling and hair morphogenesis complements the information on stem cells and helps to better understand the location of stem cells and their role in the different parts of the hair follicle cycle.
  11. Answer to comment of revisor: “Too many and unnecessary places are marked with (-)”. It may be an error in your Word formatting. There is no (-) in the submitted document and no other reviewer has notified me of this issue.
  12. Answer to comment of revisor: “Page 3: In AGA, the follicle miniaturization is accompanied by a decrease of the percentage of HFs in anagen and an increase in the telogen”. “Versican is only mentioned while explaining the miniaturization part of AGA. There are many various factors, and since versican is expressed only in DP, expression is inevitably weakened during the telogen period when the size of DP is greatly reduced. Also, I wonder why this is included in the hair cycle section”. The authors do not consider that the above information needs to be edited. If the reviewer considers that it should be in another section of the text, please notify us so that we can consider moving it to another section.
  13. Answer to comment of revisor: “Hair Follicle Stem Cells (HFSCs). Only mouse cells are mentioned”. The paragraph discusses HFSCs in general and their function, which is equivalent in both mice and humans. There is no mention of mice in the section. If the reviewer thinks the information in that section should be expanded, please let us know.
  14. Answer to comment of revisor: “Bioengineering strategies for human HF regeneration, lack of strategy content about human cells” Section 4 summarises the processes of human hair follicle tissue engineering, which is then expanded upon in the subsequent paragraphs of that section.

Reviewer 3 Report

The paper by Molina et al focused on different approaches involved in the treatement of androgenetic alopecia. In particular, authors describe therapeutics currently used including topical and oral drugs, as well as follicular unit micro-transplantation techniques. The manuscript show in a very clear manner not only the anatomy and the regeneration process of HF, but also the TE strategies to treat alopecia. 

Minor comments: 

  1. Regarding the micrografts applications, I suggest to cite also "Ruiz RG, Rosell JMC, Ceccarelli G, De Sio C, De Angelis GC, Pinto H, Astarita C, Graziano A. Progenitor-cell-enriched micrografts as a novel option for the management of androgenetic alopecia. J Cell Physiol. 2020 May;235(5):4587-4593. doi: 10.1002/jcp.29335. Epub 2019 Oct 23. PMID: 31643084" in order to give a complete overview of the applications with the Rigenera method.
  2. Please add a little paragraph and some references about the use of three-dimensional (3D) DP spheroids (DPS) for hair regeneration. The combination of 3D bioprinting and organoid model system have the potential to provide significant insights into the underlying mechanisms of HF morphogenesis. 

Author Response

Thank you for your prompt and kind reply. A new cover letter is sent to inform and justify the modifications made by the authors to the article following the reviewer's suggestions. 

  1. In section 5, a text has been added concerning the proposed article (Ruiz, R. G. et al. Progenitor-cell-enriched micrografts as a novel option for the management of androgenetic alopecia. Cell. Physiol. 235, 4587–4593 (2020).
  2. A small paragraph has been added to section 4.1 (culture) regarding 3D spheroid cultures to supplement previous information.

Best regards.

JM Llamas-Molina

Reviewer 4 Report

In the paper entitled “Tissue Engineering and Regeneration of the Human Hair Follicle in Androgenetic Alopecia. Literature Review “, the authors discuss the current and emerging hair follicle bioengineering strategies for the treatment of Androgenetic alopecia (AGA). Up to date, treatments for AGA do not always achieve a satisfactory result for the patients and may cause side effects that lead to discontinuation. Tissue engineering is one of the possible future solutions to AGA, via developing fully functional hair follicles. However, currently, the main drawbacks of the available techniques are the high costs, complexity and the difficulty of routinely translating them into clinical practice.

The review is well-structured and comprehensive. A minor spell- and grammar-check is required. Tables and figures increase the value of the manuscript.  

Author Response

Thank you for your prompt and kind reply.

Modifications that have been made are shown in red in document “Edited after major revision (changes in red). Tissue Engineering and AGA. Life Journal. Llamas-Molina et al.”. 

After the appropriate changes have been made and assessed by the reviewers, the text will be checked by a native English speaker so that the appropriate changes in grammar can be made.

Best regards,

Jose M Llamas-Molina

Round 2

Reviewer 2 Report

The author made a number of corrections, and answered appropriately.